# Basal complex: a smart wing component for automatic shape morphing

Sepehr H. Eraghi[1], Arman Toofani [1], Ramin J. A. Guilani[1,2], Shayan Ramezanpour[1], Nienke N. Bijma [3], Alireza Sedaghat[4], Armin Yasamandaryaei[4], Stanislav Gorb [3] & Hamed Rajabi [1,5 ✉]

Insect wings are adaptive structures that automatically respond to flight forces, surpassing even cutting-edge engineering shape-morphing systems. A widely accepted but not yet explicitly tested hypothesis is that a 3D component in the wing's proximal region, known as basal complex, determines the quality of wing shape changes in flight. Through our study, we validate this hypothesis, demonstrating that the basal complex plays a crucial role in both the quality and quantity of wing deformations. Systematic variations of geometric parameters of the basal complex in a set of numerical models suggest that the wings have undergone adaptations to reach maximum camber under loading. Inspired by the design of the basal complex, we develop a shape-morphing mechanism that can facilitate the shape change of morphing blades for wind turbines. This research enhances our understanding of insect wing biomechanics and provides insights for the development of simplified engineering shape-morphing systems.

[1] Mechanical Intelligence (MI) Research Group, South Bank Applied BioEngineering Research (SABER), School of Engineering, London South Bank University, London, UK. [2] Faculty of Mechanical Engineering, University of Guilan, Rasht, Iran. [3] Functional Morphology and Biomechanics, Institute of Zoology, Kiel University, Kiel, Germany. [4] Department of Mechanical Engineering, Lahijan Branch, Islamic Azad University, Lahijan, Iran. [5] Division of Mechanical Engineering and Design, School of Engineering, London South Bank University, London, UK. ✉email: rajabijh@lsbu.ac.uk

Birds and bats possess flight muscles that actively control their wing movements and deformations, thereby enhancing their flight performance. On the contrary, insect wings lack muscles, except those that are situated in the thorax and control wing movements in the basal region of the wings. Instead, insect wings consist of structural components that enable them to automatically respond to flight forces[1–6]. The controlled responses include bending, twisting and camber formation for efficient lift and thrust generation[2,7,8]. Although the direct flight muscles in the thorax can change and control the wing base profile to a limited extent, the lack of muscles within insect wings necessitates automatic shape control of the wings beyond the wing base, encoded in the wing structural design and material composition[9]. This distinguishing feature of insect wings, i.e., automatic shape control, makes them unique among all natural and engineering systems and, more importantly, a potential candidate for engineering applications that seek to achieve automatic shape control[10,11].

Dragonflies and damselflies from the insect order Odonata outperform almost any other insect in terms of flight performance. They exhibit intriguingly sophisticated flight thanks to their highly specialised wings[1–9]. Many of the wing features, including gradients of material properties[12,13] and thickness[14,15], venation pattern[2,16,17], corrugation[18,19], nodus[16,20], pterostigma[21], vein joints and joint-associated spikes[22,23], resilin patches[24–26], vein ultrastructure[23,27], flexion lines[28], and the basal complex[4,29,30] contribute to the automatic deformability of odonatan wings, and in particular, wing camber formation. A widely accepted but not yet explicitly tested hypothesis is that the basal complex—a 3D structure at the wing base with a special arrangement of veins—is key to determining the quality of wing deformations in odonatan species[2,16,29]. Although the shape, dimension, and position of the basal complex within the wing, comprising a large part of the wing's proximal region, suggest that this may be a reasonable hypothesis, the literature data are mostly descriptive, and quantitative and/or systematic investigations of the role of the basal complex in wing deformations are still rare[4,31,32]. A comprehensive study that establishes a link between the structure, material and mechanical performance of the basal complex can help us fill this literature gap. This is the overarching aim of this study.

Here we selectively collected three species of Odonata, including *Ischnura elegans* (Coenagrionidae), *Calopteryx splendens* (Calopterygidae), and *Sympetrum vulgatum* (Libellulidae) with morphologically different basal complexes and flight styles. We used a combination of experimental methods and imaging techniques, including scanning electron microscopy (SEM), micro-computed tomography (micro-CT), confocal laser scanning microscopy (CLSM), wide-field fluorescence microscopy (WFM), mechanical testing, finite element analysis (FEA), parametric modelling, conceptual design, and 3D printing to (1) examine both the structure and material of the basal complex, (2) characterise how they influence the mechanical behaviour of the basal complex, (3) determine the role of the basal complex in wing deformations, and (4) use wing-inspired design concepts in a real-world application. Our results are significant as they not only enhance our understanding of the biomechanics of insect wings but also inform the design of shape-morphing structures that do not require complicated active controls.

## Results and discussion
### Structural, material, and mechanical characterisation.
The basal complex is a 3D corrugated structure at the wing base of Odonata wings, which can comprise multiple structural elements, including arculus, discoidal cell, triangle, subtriangle,

mediocubital bar, composite vein, and a network of interconnected veins (Fig. 1A–D). Although the structural design of the basal complexes from different wings shares some commonalities, they are morphologically diverse, and their design varies from one species to another. We used SEM, CLSM, WFM, and micro-CT to investigate the structure and material properties of the basal complex of the forewings in the two damselflies, *I. elegans* and *C. splendens*, and the fore- and hindwings of the dragonfly *S. vulgatum*, as three representatives of the order Odonata (Fig. 1, Supplementary Fig. S1, Supplementary Videos S1–S4). It is important to note that the fore- and hindwings of damselflies (Zygoptera) are almost identical, whereas they show rather strong differences in dragonflies (Anisoptera).

The basal complexes of the investigated species are corrugated structures with a multitude of vein joints, resilin patches, and a network of veins and the membranes between them, comprising a large portion of the wing base (Fig. 1, Supplementary Fig. S1). They can be distinguished among the species based on their structure and material composition. The basal complex of the small damselfly *I. elegans* is the simplest in terms of design; it consists of an arculus and a discoidal cell along with the smallest network of veins (Fig. 1B). Its dorsal-ventral corrugation patterns, as well as dorsal-ventral distribution of resilin patches, are asymmetric (Supplementary Fig. S3). The basal complex of the large damselfly *C. splendens* has also a relatively simple design but possesses more structural elements than that of *I. elegans*. It includes a mediocubital bar, a composite vein, and a dense network of veins (Fig. 1C). In contrast to that observed in the wings of *I. elegans*, both the dorsal-ventral corrugation patterns and the dorsal-ventral distribution of resilin patches are almost symmetric (Supplementary Fig. S5A, Q). The dragonfly *S. vulgatum* possesses the most structurally complex basal complex among the examined species (Fig. 1A, B). It has an arculus and a discoidal cell, which is divided into two subelements called triangle and subtriangle, both radically different in fore- and hindwings. The corrugation patterns and distribution of the resilin joints show dorsoventral asymmetry. Further, the wings of *S. vulgatum* have a higher abundance of double-flexible joints in comparison to both damselfly species (Fig. 2Ai–Aviii, 2Bi–Bviii, Supplementary Figs. S2 and S4).

The differences observed in the structural and material properties of the basal complex between the studied species raise an interesting question: How do these differences influence the mechanical behaviour of the basal complex and the whole wing? To address this question, we reproduced the deformations of the wings in flight by subjecting them to a point force applied to the radial vein posterior (RP) from below (Supplementary Fig. S21), as this loading can result in deformations similar to those exhibited by the wings in flight[29]. We performed the mechanical testing on the forewings of damselflies *I. elegans* and *C. splendens* and the fore- and hindwing of the dragonfly species *S. vulgatum*. All experiments were conducted on live specimens, as insect wings are living systems, which contain haemolymph inside their veins, and their properties would change upon removal from the body[33,34]. The wings were fixed at their base using bee wax to avoid the influence of the deformations at the wing hinge on the results. We subdivided the wings into two groups of intact and cut wings. The cut wings had about one-third of the wing from the base, and the rest of the wing was removed. Testing the two groups of wings enabled us to investigate the mechanical behaviour of the basal complex both within the wing and in isolation.

The first interesting observation was that the force required to deflect the wings for 10% of the lever arm (i.e., the distance between the applied force and the fixation site) raised almost linearly with

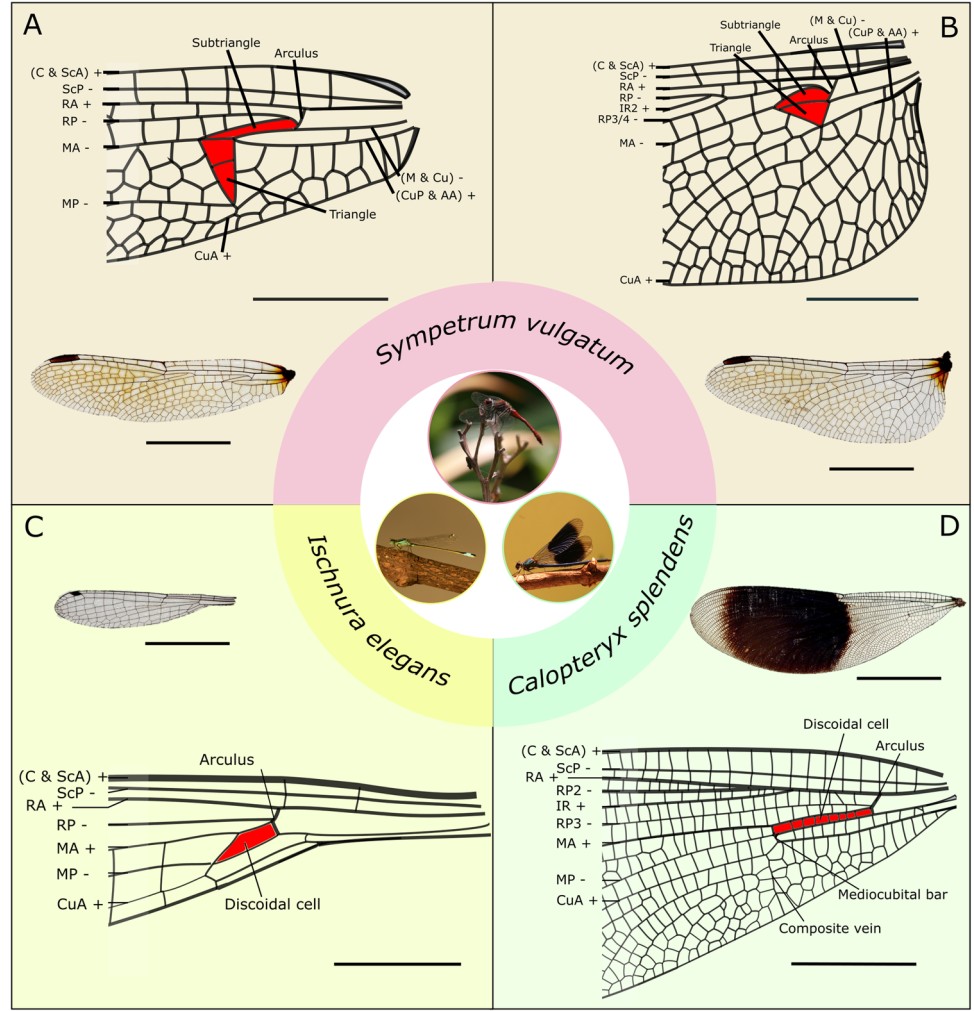

**Fig. 1 Basal complex and its key components in odonate wings. A–D** Basal complex of the dragonfly *S. vulgatum* forewing (**A**) and hindwing (**B**), the damselfly *I. elegans* forewing (**C**) and the damselfly *C. splendens* forewing (**D**) (according to the nomenclature of Rehn, 2003). Wings are shown from the dorsal side with "+" and "−" indicating if veins are raised (hill) or lowered (valley) in reference to the midline of the wing. Red areas show the discoidal cells. AA anal vein anterior, C costal vein, CuA cubital vein anterior, CuP cubital vein posterior, IR intercalated vein, MA median vein anterior, MP median vein posterior, RA radial vein anterior, RP radial vein posterior, ScA subcostal vein anterior, ScP subcostal vein posterior. Scale bars: 0.5 cm (**A**, **B**, **D**), 0.2 cm (**C**).

the displacement (Fig. 2Ci, Supplementary Figs. S3–S5). This indicates the linear response of the wing to forces/displacements applied within this range. Second, applying a force to the dorsal side of the forewing of *S. vulgatum*, as an example, caused only a small deformation in the basal complex (Fig. 2Ciii, Cv). In contrast, a force applied to the ventral side resulted in a noticeable deformation in the basal complex, which also formed a cambered section (Fig. 2Cvii, Cix). Comparison of the results from the intact and shortened wings showed that regardless of the direction of the applied force, the deformation patterns of the intact wings and cut wings were similar (Fig. 2Ciii–Cx, Supplementary Figs. S3–S5). In both dorsal and ventral loadings, the deformations observed in the whole wings were influenced by those of the basal complex— applying a force on the ventral side of the wings resulted in a camber formation (Fig. 2Cvi, Cx). The only difference was that the camber formation in cut wings was slightly shifted towards the base of the wing, an effect that can be due to the absence of the pterostigma in the cut wings as a counterweight at the wing tip[19]. These results suggest that the two-thirds of the wing domain distal to the basal complex may have a comparatively small influence on the wing deformation pattern, a finding which supports the hypothesis mentioned earlier.

The mechanism of deformation is relatively simple yet very effective (Fig. 2Di–Diii). When the force is applied to the radial vein posterior (RP) from the ventral side, RP raises and elevates the radial vein anterior (RA). The elevation of the radial vein anterior (RA) causes the leading-edge spar to move downwards slightly. The rotation of the wing about the axis of the median vein anterior (MA) lowers the trailing edge. This deformation is accompanied by the rotation of the triangle and subtriangle, promoting the camber. The mechanism of deformation is almost the same between the investigated wings and similar to that described previously for comparatively simple Diptera wings[31] (see Supplementary Note 1 for more information).

An interesting finding is that the wings of both the damselfly *I. elegans* and the dragonfly *S. vulgatum* (Fig. 2C, Supplementary Figs. S3 and S4) showed asymmetric deformation patterns when bent upwards and downwards, whereas those of the damselfly *C. splendens* exhibited an almost symmetric deformation (Supplementary Figs. S5 and S10). Specifically, for the wings of *S. vulgatum*, although no significant difference was found in the bending moments required to deflect the wings between the two opposite directions, a pressure applied to the wings resulted in camber formation only when they were loaded on the ventral side

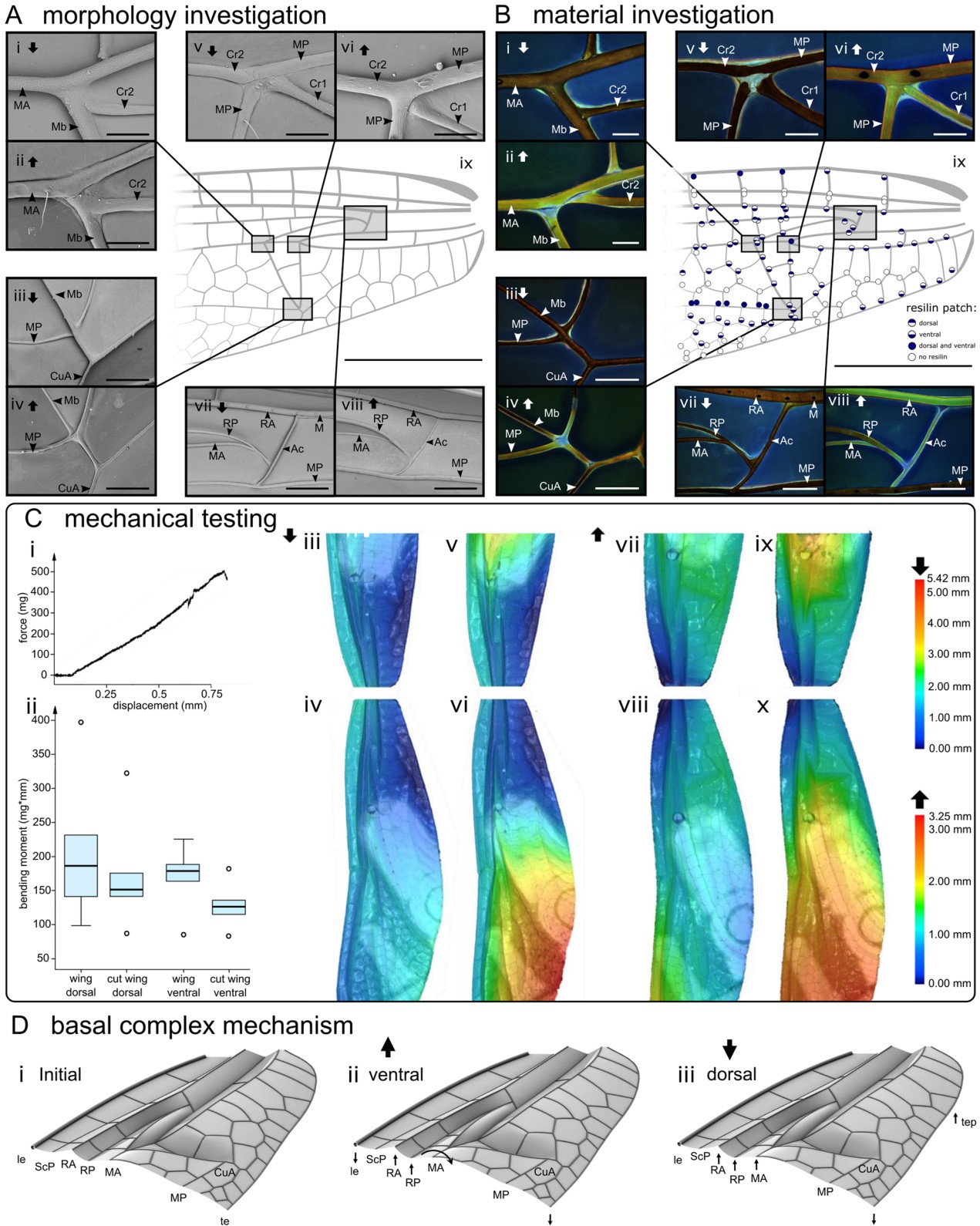

**A  morphology investigation**

**B  material investigation**

**C  mechanical testing**

**D  basal complex mechanism**

i Initial    ii ventral    iii dorsal

(Fig. 2Cii–Cx, Supplementary Fig. S3). This is the characteristic asymmetric deformation pattern of dragonfly wings, which generates lift in the downstrokes[29]. This dorsal-ventral asymmetry is likely the result of the distribution of asymmetric joints and pre-cambered cross-sections of the wings (Fig. 2B, Supplementary Fig. S2), which are less obvious for wings of *C. splendens* (Supplementary Fig. S5).

To better understand how the basal complex influences wing deformations, we developed one of the most comprehensive finite element models of insect wings to date. This is a model of the forewing of the dragonfly *S. vulgatum*, which includes many morphological details of the wings, including longitudinal veins, cross veins, vein joints, layered structure of veins, resilin patches, nodus, pterostigma, corrugation patterns, and membranes

**Fig. 2 Structural, material, and mechanical characterisation of the basal complex of the forewing of the dragonfly *S. vulgatum*. A** Scanning electron microscopy (SEM) images of the forewing. SEM images of the arculus (**Avii**, **Aviii**) and the basal-anterior corner (**Av**, **Avi**), the apical-anterior corner (**Ai**, **Aii**), and the posterior corner (**Aiii**, **Aiv**) of the triangle. Venation pattern of the basal complex of the forewing (**Aix**). Dorsal and ventral sides are indicated by downward-pointing and upward-pointing arrows, respectively. **B** Confocal laser scanning microscopy (CLSM) maximum intensity projection images, showing the occurrence of resilin in the arculus (**Bvii**, **Bviii**) and the corners of the triangle: basal corner (**Bv**, **Bvi**), apical corner (**Bi**, **Bii**) and the posterior corner (**Biii**, **Biv**). The blue colour indicates the presence of resilin. Red and green colours show highly sclerotised and less sclerotised cuticles, respectively. Distribution map of resilin patches (**Bix**). The occurrence of resilin is illustrated by the blue colour according to its location in the wing (dorsal or/and ventral). White circles indicate firmly connected joints that lack resilin. The grey area illustrates the anal loop (**Aix**, **Bix**). **C** Results of mechanical testing of wing deformability. Intact and cut forewings were deflected from the dorsal and ventral sides. Representative force-displacement curve obtained from the forewing (**Ci**). Bending moments required to deflect the wings (**Cii**). The top, middle, lower lines, and dots in box-and-whisker plots are the lower extreme, median, upper extreme, and outlier data, respectively. **Ciii–Cx** Height profiles of the forewing. Intact and cut forewings were deflected from the dorsal and ventral sides. Resulting height profiles before (**Ciii**, **Civ**, **Cvii**, **Cviii**) and after (**Cv**, **Cvi**, **Cix**, **Cx**) deflection. Downward and upward-pointing arrows indicate the side from which forces were applied (dorsal side: **Ciii–Cvi** and ventral side: **Cvii-Cx**). **D** Mechanism of deformation of the basal complex (**Di**) when subjected to forces on the ventral (**Dii**) and the dorsal sides (**Diii**). Ac arculus, Cr cross vein, Cu cubital vein, CuA cubital vein anterior, M median vein, MA median vein anterior, Mb mediocubital bar, MP median vein posterior, RA radial vein anterior, RP radial vein posterior. Scale bars: 0.5 mm (**Aix**, **Bix**), 100 μm (**Ai**, **Aii**, **Av**, **Avi**, **Bi**, **Bii**, **Bv**, **Bvi**), 250 μm (**Aiii**, **Aiv**, **Biii**, **Biv**), 500 μm (**Avii**, **Aviii**, **Bvii**, **Bviii**).

(Supplementary Fig. S11). We used this model to investigate the influence of the basal complex on both the quality and quantity of wing bending, twisting, and camber. After verifying the validity of the model and its accuracy in simulating wing response to loadings (see "Methods" and Supplementary Fig. S12), we removed the basal complex by modelling it as a flat plate. We found that the removal of the basal complex from the wing noticeably altered its deformations; wing bending increased by 1.7 times, the wing twisted in the opposite direction, and wing camber decreased by 1.7 times (Fig. 3A, 3Bi–Bii, Supplementary Fig. S13). Considering the detrimental effect of the wing bending and the importance of the wing twisting and camber for the aerodynamic load generation, the results suggest the key role of the basal complex in the automatic shape changes of the wings, supporting our experiments on the real wings.

We used our comprehensive numerical model in a comparative study to understand how changing the geometric parameters of the basal complex, including those shown in Fig. 3Ci, Cii, can influence wing camber formation. Except for the rotation angle δ of the longitudinal veins, which shows a linear decreasing trend by increasing the angle, the variation of other geometric parameters resulted in non-linear changes in the wing camber. Among the studied parameters, the parameter 'longitudinal veins rotation angle δ' had the strongest influence; changing the angle by 16 degrees increased the camber by 1.4 times (Fig. 3C, Supplementary Figs. S14–S19). Subtriangle rotation angle γ, on the other hand, had the smallest influence on the wing camber. Interestingly, the basal complex model with geometric parameters as those of the basal complex of the forewing of *S. vulgatum* showed one of the largest cambers among all model variations (Fig. 3C). This is important as it suggests the adaptation of the wing basal complex to develop cambered sections in flight, a phenomenon that can increase the ability of the wings to produce aerodynamic forces.

**Bioinspired design and application.** The design strategies of the basal complex investigated here demonstrated that the basal complex is an automatic mechanism for shape morphing, which can change configuration upon loading to form a cambered shape. The basal complex indeed represents a striking example that can inform studies in mechanical intelligence (MI), a new research area that exploits nature-inspired mechanisms for automatic adaptability and applies them to the design of structural components[35]. In general, this mechanism can inspire the design of shape-morphing structures that do not require complicated actuations to achieve functionality. In particular, the basal complex can enable us to develop bioinspired symmetric

and asymmetric shape-changing mechanisms for a variety of high-tech applications, such as flapping-wing robots, shape-adaptive turbine blades, shape-adaptive airplane wings, airplane wing flaps, and shape-adaptive car spoilers.

Using the conceptual parametric design, here we developed a simplified model inspired by the wing basal complex. The bioinspired mechanism consists of links that are connected to each other via shafts and pin joints (Fig. 4Ai, Supplementary Fig. S21). To guide the link motions, we designed a frame which included a few pin slot joints (Fig. 4Aii, Supplementary Fig. S21). By applying an upward force on the link RP, the mid-part of the mechanism moves upwards, whereas the trailing edge is pushed down, thereby creating a cambered configuration similar to that seen in the natural wings (Fig. 4Aiii, Aiv, Supplementary Video S5). Using a 3D-printed prototype, we tested the functionality of our bioinspired design in practice (Fig. 4Bi–Bii). This was found to be a simple yet functional mechanism that can be easily manufactured and simply assembled/disassembled. Using the example of deformable blades of a wind turbine, we showed in a conceptual design that the mechanism is potentially capable of automatic shape control of the chord-wise camber of the blades and their angle of attack under varying wind forces, thus enhancing their efficiency (Fig. 4C, 4Di–Diii)[36–40]. Our proposed shape-morphing mechanism can offer advantages over the conventional wind turbine blades, which only work efficiently at specific operating points or over the shape-morphing blades that require complex motor control systems and/or smart material design[36–40].

## Methods

**Specimens.** In this study, we used adult damselflies *Ischnura elegans* (Vander Linden, 1820) (Coenagrionidae) and *Calopteryx splendens* (Harris, 1782) (Calopterygidae) and adult dragonflies *Sympetrum vulgatum* (Linnaeus, 1758) (Libellulidae). These species were used as they have different wing morphologies: *I. elegans* has small wings with relatively simple architecture, *C. splendens* is a large damselfly with a dense network of reinforcing veins, and *S. vulgatum* is a mid-size dragonfly with fore and hindwings that differ in venation and shape and in contrast to the other two damselfly species has broad-based wings. The specimens used for this study were collected in Kiel, Germany (*I. elegans*, *C. splendens* and *S. vulgatum*) in 2016 and in Crimea, Ukraine, in 1997 (*C. splendens*).

For mechanical testing, we used only fresh wings of male individuals of *I. elegans* and *S. vulgatum* that were caught with the permission of the Landesamt für Natur und Umwelt des Landes Schleswig-Holstein (LANU). We used fresh specimens because the properties of wings can change upon desiccation. Prior to the experiment, the insects were anaesthetised using $CO_2$.

### Morphology investigation

*Scanning electron microscopy (SEM).* The basal complex of the wings was examined with SEM. Prior to the SEM, air-dried wings of the male specimen were sputter coated with a 10 nm gold-palladium layer using a high vacuum sputter coater (Leica SCD 500, Leica Microsystems GmbH, Wetzlar, Germany). For the SEM, a

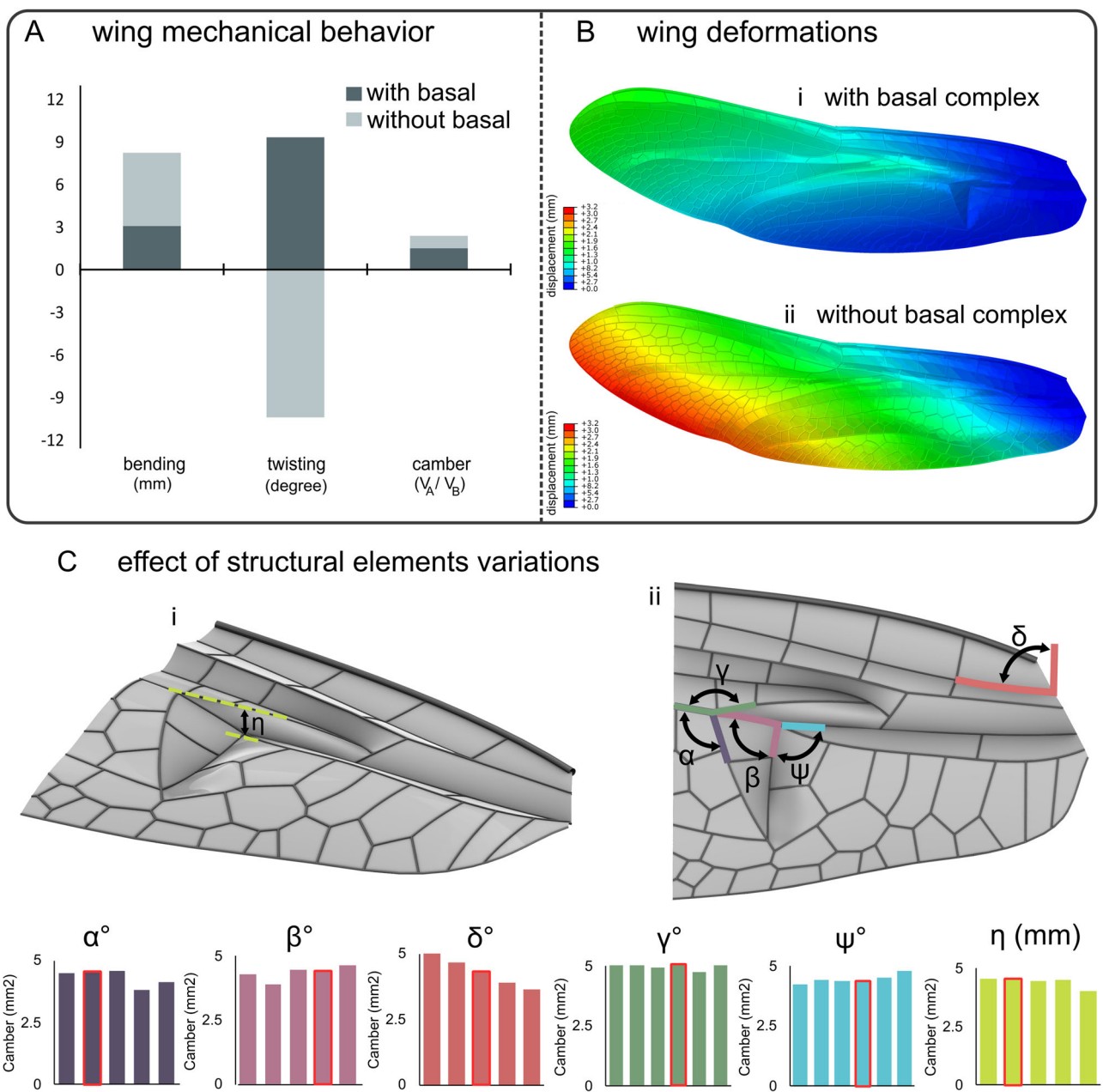

**Fig. 3 Finite element analysis of parametric models of the basal complex developed based on the forewing of *Sympetrum vulgatum*. A** Bending, twisting, and camber in wing models with and without basal complex. $V_B$ and $V_A$ refer to the volume under the wing before and after deformation. **B** Deformation of the wing with and without basal complex from the top view. **C** Effect of variations of the geometric parameters of the basal complex on camber formation. The panel shows the perspective view (**Ci**) and the top view (**Cii**) of the basal complex. Altered geometric parameters include triangle exterior angle ($\alpha$), triangle interior angle ($\beta$), longitudinal veins rotation angle ($\delta$), triangle rotation angle ($\psi$), subtriangle rotation angle ($\gamma$), triangle incline depth ($\eta$). The angles measured in the real wings are highlighted using a red outline.

Hitachi TM3000 Tabletop Microscope (Hitachi High-Tech. Corp., Tokyo, Japan) at an accelerating voltage of 15 kV and a magnification of 100X–600X was used.

*Micro-computed tomography (micro-CT).* To investigate the morphology of the basal complex of Odonata wings, we used micro-CT scanning. For this, air-dried wings were scanned using a Skyscan 1172 desktop micro-CT scanner (Bruker micro-CT, Kontich, Belgium). Due to the similarity of fore- and hindwings in damselflies, only the forewings of *I. elegans* and *C. splendens* were examined. The specimens were scanned at a source voltage of 35–40 kV and a source current of 216–250 µA (Supplementary Table S1). For 3D reconstruction of the basal complex, we used NRecon (Bruker micro-CT, Kontich, Belgium) and generated images with pixel sizes of 2.67–5.00 µm. Processing and visualisation were done with Amira 6.0.1 (FEI Visualization Science Group, Berlin, Germany).

**Material composition investigation**

*Confocal laser scanning microscopy (CLSM).* Wings of male and female specimens were examined with a CLSM, which enabled us to characterise the differences in the material composition based on their different autofluorescence characteristics[41]. The elastomeric protein resilin has a very narrow wavelength band of around 420 nm, in which it emits blue autofluorescence (Andersen and Weis-Fogh 1964), allowing the identification of resilin-dominating areas (Supplementary Figs. S6–S9).

For CLSM, air-dried wings were rehydrated for 24 h in a 10:1 mix of distilled water and phosphate-buffered saline (PBS). Following short immersion in 70% ethanol, the wings were embedded in glycerine (≥99.5% free of water) on a glass slide and covered by a cover slip. To prevent direct contact between the wings and cover slips, five reinforcement rings (used for office work) were attached to the glass slide. The wings were visualised with a CLSM Zeiss LSM 700 (Carl Zeiss

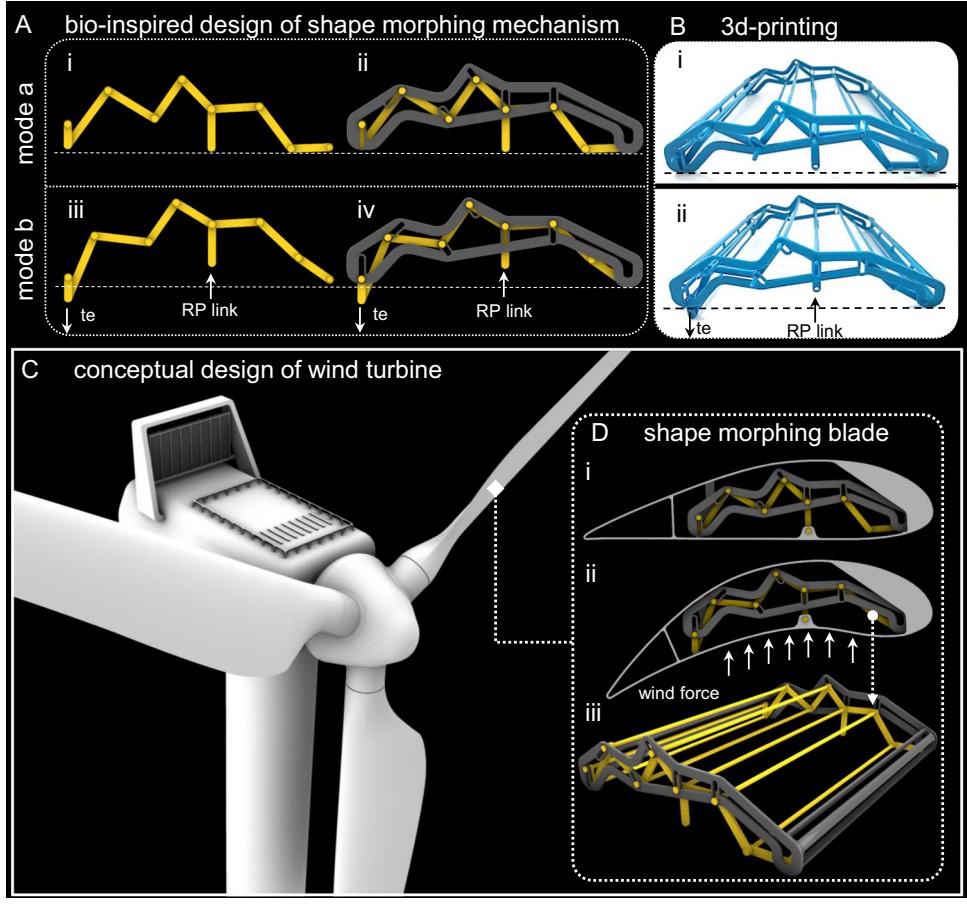

**Fig. 4 Bioinspired automatic shape morphing. A** Bioinspired shape-morphing mechanism with the ability to change its configuration between mode a (**Ai**, **Aii**) and mode b (**Aiii**, **Aiv**). **B** 3D printing and testing. 3D-printed mechanism in mode a (**Bi**), in mode b (**Bii**). **C** Conceptual turbine blade.
**D** Application of the bioinspired mechanism in shape-morphing turbine blade in mode a (initial configuration) (**Di**) and in mode b (cambered configuration) (**Dii**). **Diii** A perspective view of the mechanism. te trailing edge.

MicroImaging GmbH) equipped with an upright microscope (Zeiss Axio Imager.M1m). Four lasers with 405, 488, 555, and 639 nm wavelengths were used during scanning. A bandpass emission filter transmitting 420–480 nm and three longpass emission filters with ≥490, ≥560 and ≥640 nm were used to collect emitted light from the specimens. The gain and intensity for each laser were manually adjusted to avoid oversaturation. Samples were excited sequentially with the four lasers, and images were collected with a line average equal to two. The pinhole size was set to 1 AU (Airy Unit).

Overlapping optical image stacks were created for the entire thickness of the specimen. Maximum intensity projections were created based on the collected image stacks using ZEN 2009 software, where colours represent certain detected wavelengths (blue for 420–480 nm, green for ≥560 nm, and red for ≥560 and ≥640 nm).

*Wide-field fluorescence microscopy (WFM).* To test the distribution of resilin in vein joints, as a complementary method to the CLSM, we used fluorescence microscopy (FM). For this purpose, we examined specimens using a Zeiss Axioplan fluorescence microscope (Carl Zeiss Microscopy GmbH, Germany). The preparation method is the same as that conducted for the CLSM. For visualisation of the autofluorescence of resilin, the microscope was equipped with a DAPI filter set with a bandpass excitation filter transmitting 321–378 nm and a bandpass emission filter transmitting 420–470 nm. Light intensity was manually adjusted to obtain optimal saturation. Images were taken using a Zeiss Axio Cam Mrc (Carl Zeiss Microscopy GmbH, Germany) and the Zeiss Efficient Navigation software (Carl Zeiss Micro-Imaging GmbH). Wings were examined from both the dorsal and ventral sides.

### Investigation of the mechanical behaviour

*Mechanical testing.* Experiments were carried out to characterise the deformation of the wings under loading. For each specimen, we recorded forces, displacements, and 3D deformation patterns of the wings. Freshly killed five male individuals of each of the species *I. elegans*, *C. splendens* and *S. vulgatum* were used in experiments. In order to prevent desiccation, which can change the material properties, and thereby the mechanical behaviour of the wings, the tests were performed on

live insects, and a droplet of bee wax was used to fix the wings at their joints to the body (Supplementary Fig. S22).

For force measurements during wing deflection, we used a 10 g force sensor (WPI Fort 10g, World Precision Instruments, Florida, US) connected to an MP 100 data acquisition system (Biopac System Inc, Goleta, CA) (Supplementary Fig. S22). The data acquisition rate was set to 2000 Hz. The wings were deflected by applying a point force to the radial vein posterior (RP) at ~30% of the wing length away from the wing base (Supplementary Fig. S22). This is expected to reproduce the deformation of the wings in flight[29]. The force was applied using the blunt side of an insect pin attached to the force sensor. The displacements of the force sensor were precisely controlled using a micromanipulator (Physik Instrumente (PI) GmbH & Co. KG, Karlsruhe, Germany) via the software PIMikroMover 2.4.4.6. The displacement speed was set as 0.05 mm/s. A displacement equal to 10% of the wing's length was applied to each specimen. The data from the data acquisition system was recorded using the software AcqKnowledge 3.7 over the course of the loading. The experiment was conducted under the 3D measurement microscope Keyence VR300 (Keyence Microscope Europe, Mechelen, Belgium), which enabled us to capture the 3D profile of the wings before and after loading. The experiments were performed on both the dorsal and ventral sides of the wings.

The testing procedure was repeated for wings in which two-thirds of the wing length was removed from the distal part (i.e., only the basal complex was left). This enabled us to compare the deformations of the intact and cut wings and characterise the role of the basal complex on wing deformations. The testing of each specimen never exceeded 2 h.

*Finite element analysis (FEA).* We used finite element analysis to investigate the influence of the basal complex and its geometric parameters on the mechanical response of the wings to loading. For this purpose, we developed a detailed 3D model of the forewing of the dragonfly *S. vulgatum* using the computer-aided design software CATIA v5 (Dassault Systèmes, Sureness, France) based on the data from micro-CT and SEM. In this model, the key structural features of wings, including the corrugations, pre-camber, thickness gradient, longitudinal veins, cross veins, the basal complex, resilin joints and resilin-rich layers within the veins, were modelled. The thickness gradient included changes in both the thickness of

the membranes and the costal vein. We developed this gradient using data reported by Jongerius and Lentink[14]. Specifically, we set the thickness of the membrane to change from 20 to 5 μm. In the thickest areas, we set the radius and thickness of the costal vein to be 80 μm and 40 μm, respectively, and in the thinnest parts, we set them to 50 μm and 20 μm, respectively. We also made a few simplifying assumptions by considering the veins (except the costal vein) as circular tubes with a constant radius and thickness of 20 μm and 7 μm, respectively, throughout the model. Resilin patches were modelled through connections of different modelled materials in the assembly, regardless of whether they were on the dorsal or the ventral side.

To quantify the role of the basal complex on wing deformability, we developed another model by removing the basal complex from the wing. In other words, the second model is the same as our reference model, except that the basal part of the model is flat.

Next, by changing the geometric parameters of the model of the *S. vulgatum* forewing, we developed 27 models with distinct morphologies of the basal complex for a comparative study. The models, in which the geometric parameters were changed at set intervals, enabled us to quantify the effect of the geometric parameters of the basal complex on its camber formation. The geometric parameters included in our study were triangle exterior angle ($\alpha$), triangle interior angle ($\beta$), subtriangle rotation angle ($\gamma$), longitudinal veins rotation angle ($\delta$), triangle incline depth ($\eta$), and triangle rotation angle ($\psi$) (Fig. 3C). For complete wing models, we utilised the volume below the wings to characterise camber. For cut wing models (models of the basal complex only), we characterised the camber by calculating the area below the free end of the models, as it was easily accessible (Supplementary Fig. S20).

For simulations, the models were imported into the ABAQUS v.6.14 FE software package (Simulia, Providence, RI). The Young's modulus and the Poisson's ratio were set as 3 GPa and 0.49 for veins and 1.86 GPa and 0.49 for membranes[42,43], respectively. The Young's modulus and the Poisson's ratio of resilin were 2 MPa and 0.49[44], respectively (Supplementary Fig. S11). The two-node beam elements B31 and the general-purpose shell elements S4R were used to model veins and membranes, respectively. The models were subjected to the same loading and boundary conditions as those used in mechanical testing described earlier. Specifically, the wings were fixed at their base (both displacements and rotations) and then subjected to a displacement at RP from the dorsal/ventral sides. A mesh convergence analysis was conducted to find the suitable mesh size resulting in reasonably accurate results in each simulation. We selected the mesh size at which the results no longer changed significantly as the appropriate mesh size for our models.

To validate the FEA model, we compared the force-displacement curves obtained from the experiments conducted on intact wings with their respective simulations (Supplementary Fig. S12). This comparison allowed us to assess the accuracy of the model's predictions. We measured the average stiffness of both the wings and their corresponding FEA models from the two sets of curves. Our validation results revealed a difference of ~12% between the measured stiffness values.

### Bioinspired application

*Parametric modelling and conceptual design.* Using the computer-aided design software (Rhinoceros 7 3D, Seattle, WA) and Grasshopper plugin, we developed a parametric model of a mechanism for camber formation inspired by the basal complex. The mechanism consists of 16 linkages, 10 shafts, and 20 shaft pins, which are assembled within a frame. The dimensions of the elements of the mechanism are set as variable parameters and can be tuned to satisfy the requirements of various design purposes. The mechanism was designed to undergo considerable deformations and form a cambered shape when subject to upward/downward forces on the RP linkage (Fig. 4, Supplementary Fig. S21). We further developed a conceptual model of a deformable wind turbine blade to demonstrate the application of our bioinspired mechanism in a real-world engineering system (Fig. 4).

*Fabrication.* To show the functionality of the bioinspired camber formation mechanism in practice, the individual elements of the mechanism were first fabricated using 3D printing and then assembled. To this end, we used a Creality Ender 3 FDM 3D Printer (Shenzhen Creality 3D Technology Co Ltd., Shenzhen, China) and polylactic acid (PLA) filament (Basicfil filament, filament diameter of 1.75 mm). Printing was done at a nozzle temperature of 220 °C, a bed temperature of 60 °C, and a print speed of 40 mm/s. The layer height was set at 0.2 mm. The parts were printed at 100% infill density. No postprocessing was performed on the 3D-printed parts.

*Statistics and reproducibility.* Statistical analyses were performed to compare the bending moments required to deform the wings at different states. Specifically, we used *t*-test to compare the bending moments between the dorsal and ventral side loadings. We also used analysis of variance (ANOVA) to compare the bending moments between intact and cut wings.

**Reporting summary**. Further information on research design is available in the Nature Portfolio Reporting Summary linked to this article.

## Data availability

The authors declare that the data supporting the findings of this study are available within the paper and its supplementary information files. Specifically, the data from mechanical testing and numerical simulations can be found in Supplementary Table S2 and Supplementary Figs. S14–19, respectively.

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

## Author contributions

Conceptualisation: S.G., H.R.; Supervision: S.G., H.R., A.S.; Formal analysis: S.R., N.N.B., H.R.; Investigation—modelling and simulation: S.R., R.J.A.G.; Investigation—microscopy: N.N.B., H.R.; Investigation—product design: S.H.E., A.T.; Investigation—mechanical testing: N.N.B., H.R.; Investigation—3D printing: A.Y.; Project administration: H.R.; Resources: S.G., H.R.; Methodology: S.G., H.R.; Validation: S.R., R.J.A.G.; Visualisation: A.T., S.H.E., N.N.B.; Writing—original draft preparation: S.H.E., N.N.B., H.R.; Writing—review and editing: S.H.E., A.T., A.S., S.G., H.R.

## Competing interests

The authors declare no competing interests.
