## [Peer Review File · Communications Biology]

Reviewers' comments:

Reviewer #1 (Remarks to the Author):

This in-depth study investigates the basal complex in high performance insect fliers, dragonflies and damselflies. The authors focused on a structurally essential portion of the wing, that has been overlooked, a triangular region near the wing base called the basal complex. Using live insects the authors measure deflection in intact and cut wings. They then specifically asked how the basal complex changes (as it deforms) the mechanical response of the wings to bending forces.

The authors developed a novel finite element model of a dragonfly wing, and were able to model the wing with and without the basal complex. This model goes beyond simple architecture, veins and membrane, but adds in material properties like resilin patches into the model's material distribution. Figure 3B is an effective and stark example of how the wing properties would shift without the basal complex. Essentially the wing bendiness increases, twisting occurs opposite, and camber decreases. This result, supports the authors' initial hypothesis, that the basal complex is necessary for wing deformation in flight — and thus perhaps could be a good candidate for bioinspired design (which the authors then model and 3D print).

Within the field of insect flight, this is the most thorough analysis of insect wing deformation, testing a unique hypothesis about the basal complex. The authors also use live insects to perform their measurements, and the majority of this field use separated, dried wings. This data uses a variety of well-known techniques, but takes it a step further in combining SEM, micro-CT, CLSM, WFM, FEA, mechanical testing, parametric modeling, and 3D printing.

What's additionally impressive is the explanation and figures in the supplement. The supplement is extremely detailed and I wish the paper could feature more of those figures as well, like the methods figure S22 and the FEM analysis with/without the basal complex in S13.

I recommend this for publication with minor edits.

General notes:

-Forewing and hindwing are one word

-Figure 2 - there is no (D) in the caption

-Page 7, first paragraph. Could you define RP in the main text? You do define it in the caption, but I had to search for what it meant. Same with RA and MA on page 8, middle paragraph.

-Further, why choose RP to load with a point force? I assume it's because it directly joins with the arculus, thus having a significant deformation on the basal complex, however it seems like MA could also do that.

-Figure S1 and Figure 1 are essentially the same, however I like S1 a little more for the main manuscript, since it shows the actual wings and the insects.

-Movies - for some reason, I do not have access to the movies - but I saw them listed at the end of the supplement. An interesting feature to this study could be a movie of your biomechanical model morphing - not necessary for publishing, just something I'd want to see (perhaps that is what Movie S5 shows?)

Reviewer #2 (Remarks to the Author):

In this work, a novel study was done to quantitatively understand the behavior of the basal complex in three species of Odonata. Mechanical testing performed on live insect wings showed an increase of camber when subject to a point force applied to the ventral side of the wing, representative of in-flight wing loading. Through several microscopy techniques, a detailed morphology investigation revealed the detailed geometry of wing-veins, chordwise corrugation,

and basal-complex topography, as well as the distribution of resilin patches, all of which were used in the development of a very detailed finite element based model of the wing. Using this model for finite element analysis, several of the geometric parameters in the basal complex were altered to understand their influence on the camber and deformation of the wing under point loading. Finally, the insights gained from the Odonata basal-complex were applied to a design of a prototyped automatic shape morphing wind-turbine blade, which is expected to increase wind energy harvesting efficiency of the blade.

The novelty in this work is found primarily in (i) its quantitative nature, where many past studies of the basal-complex have been primarily qualitative, (ii) the parameter study performed to find the relative influence that many geometric parameters within the basal-complex have on the deformed state of the wing, and (iii) a proposed idea for a bio-inspired shape morphing wind-turbine blade.

The following revisions are suggested in no particular order:

1. In Figure 1, the Scale bars described in the caption are unclear. It seems A, B, C, and D are referenced, but only images A, B, and C are shown.
2. In the 4th paragraph of section 2.1, the deflection is referenced at 10% of the lever arm, while in the second paragraph of section 3.4, a deflection of 10% of the wing's length is mentioned. It is unclear if these should be the same displacements, but they seem to reference different values as the "lever arm" refers to only 1/3 of the wing's lengths. Consider rephrasing for clarity.
3. Consider adding an explanation or graphic representation (possibly to figure 3) of how camber is being quantified in this study. Va and Vb mentioned in the caption of figure 3 refer to the volume under the wing, however this does not align with the area and units of mm² shown in S14-S19. It is possible that volume changes under the full wing are not linearly proportional to area changes under the 1/3 spanwise location.
4. Figure 4.B should be referred to as a CAD rendering of the 3D-printed part, unless it is actually a photo of the physical part.
5. The final sentence of paragraph 1 in section 3.1 implies that the specimens studied were gathered in 2016 and 1997, which is contradicted in the following sentence claiming that only "fresh" wings were used. Please correct these errors or explain specifically what the specimens from past decades were used for.
6. In section 3.4, the phrase "fix the wings at their joints" implies that the wings were removed and reattached at the joint. Consider rephrasing this sentence to explain that the bee wax was used to immobilize the wing-hinge joint.
7. Validation of the FEA model in section 3.4 should be included, possibly referencing to figure S12. If the camber area under the deformed wing (S20) was gathered experimentally and can be validated against, this would significantly strengthen the reader's trust in the modeled results.

**The following comments on the FEA model could be addressed through an appendix/supplemental section to describe the model in greater detail. As this model was used to find the major trends of the parameter study, more details of the FEA model should be explained.
**

8. In section 3.4, a thickness gradient is mentioned in the list of structural features considered in the FEA model. Does this refer to membrane thickness, vein thickness, or something else? How was this measured and applied to the model? Vein thickness is later described as constant, and a thickness distribution of the shell elements is not mentioned if it was used to represent membrane thickness gradients.
9. It is mentioned that resilin patches and resilin joints were modelled, however this was not explained. Perhaps this was through connections of different modeled materials in the assembly?
10. The application of boundary conditions in the FEA model is unclear.
11. How was the discretization/mesh verified for this FEA model?

Overall, the paper presents novel work on the basal complex, and the quantitative study through physical experimentation and computational modeling was very well performed and is an interesting contribution to the field. More information about the particular setup of the FEA model

should be included either in the body or the text or in an appendix of some sort. The 3D printed bioinspired wind-turbine structure is an interesting application of the work, however this section of the paper seems to be explained in less detail (this is not a major problem, but further discussion and quantitative evaluation of the proposed structure might strengthen this section of the paper for some readers if the authors deem such an addition appropriate).

Response to Reviewers

Reviewer #1:

This in-depth study investigates the basal complex in high performance insect fliers, dragonflies and damselflies. The authors focused on a structurally essential portion of the wing, that has been overlooked, a triangular region near the wing base called the basal complex. Using live insects the authors measure deflection in intact and cut wings. They then specifically asked how the basal complex changes (as it deforms) the mechanical response of the wings to bending forces.

The authors developed a novel finite element model of a dragonfly wing, and were able to model the wing with and without the basal complex. This model goes beyond simple architecture, veins and membrane, but adds in material properties like resilin patches into the model's material distribution. Figure 3B is an effective and stark example of how the wing properties would shift without the basal complex. Essentially the wing bendiness increases, twisting occurs opposite, and camber decreases. This result, supports the authors' initial hypothesis, that the basal complex is necessary for wing deformation in flight — and thus perhaps could be a good candidate for bioinspired design (which the authors then model and and 3D print).

Within the field of insect flight, this is the most thorough analysis of insect wing deformation, testing a unique hypothesis about the basal complex. The authors also use live insects to perform their measurements, and the majority of this field use separated, dried wings. This data uses a variety of well-known techniques, but takes it a step further in combining SEM, micro-CT, CLSM, WFM, FEA, mechanical testing, parametric modeling, and 3D printing.

What's additionally impressive is the explanation and figures in the supplement. The supplement is extremely detailed and I wish the paper could feature more of those figures as well, like the methods figure S22 and the FEM analysis with/without the basal complex in S13.

I recommend this for publication with minor edits.

Response: Thank you for your detailed and encouraging feedback on our paper. We are very pleased that you find our work to be significant and that you believe it makes a valuable contribution to the field.

We have carefully considered your comments and have made the following revisions to the manuscript: (1) We have added more detail to the methods section, including a description of the procedures used to collect and analyse the data; (2) We have updated the figures in the manuscript to better represent our results; and (3) We have made minor edits to the text to improve clarity and accuracy.

General notes:

-Forewing and hindwing are one word

Response: Corrected.

-Figure 2 - there is no (D) in the caption

Response: Apologies for this – now corrected.

-Page 7, first paragraph. Could you define RP in the main text? You do define it in the caption, but I had to search for what it meant. Same with RA and MA on page 8, middle paragraph.

Response: Absolutely! Done.

-Further, why choose RP to load with a point force? I assume it's because it directly joins with the arculus, thus having a significant deformation on the basal complex, however it seems like MA could also do that.

Response: In their study, published in “Model organisms for ecological and evolutionary research” in 2008, Wootton and Newman (2008) suggested that a point force applied to RP from below can reproduce wing deformations in real wings and also in card models of the wings. We agree with the reviewer that applying a force to MA can also result in a similar effect, and this has indeed been shown by Wootton and Newman (2008) too.

-Figure S1 and Figure 1 are essentially the same, however I like S1 a little more for the main manuscript, since it shows the actual wings and the insects.

Response: We agree with the reviewer. That is why we replaced Fig 1 with Fig S1 and made a few minor modifications.

-Movies - for some reason, I do not have access to the movies - but I saw them listed at the end of the supplement. An interesting feature to this study could be a movie of your biomechanical model morphing - not necessary for publishing, just something I'd want to see (perhaps that is what Movie S5 shows?)

Response: Thank you for pointing this out. We will make sure that the videos have been properly uploaded on the journal's platform. We have revised Video S5 to include both the model and prototype of the bioinspired shape morphing mechanism.

Reviewer #2:

Comment:

In this work, a novel study was done to quantitatively understand the behavior of the basal complex in three species of Odonata. Mechanical testing performed on live insect wings showed an increase of camber when subject to a point force applied to the ventral side of the wing, representative of in-flight wing loading. Through several microscopy techniques, a detailed morphology investigation revealed the detailed geometry of wing-veins, chordwise corrugation, and basal-complex topography, as well as the distribution of resilin patches, all of which were used in the development of a very detailed finite element based model of the wing. Using this model for finite element analysis, several of the geometric parameters in the basal complex were altered to understand their influence on the camber and deformation of the wing under point loading. Finally, the insights gained from the Odonata basal-complex were applied to a design of a prototyped automatic shape morphing wind-turbine blade, which is expected to increase wind energy harvesting efficiency of the blade.

The novelty in this work is found primarily in (i) its quantitative nature, where many past studies of the basal-complex have been primarily qualitative, (ii) the parameter study performed to find the relative influence that many geometric parameters within the basal-complex have on the deformed state of the wing, and (iii) a proposed idea for a bio-inspired shape morphing wind-turbine blade.

Response: We thank the reviewer for their encouraging feedback and acknowledging the novelty of our study. We have done our best to address their comments in our revised manuscript.

The following revisions are suggested in no particular order:

1. In Figure 1, the Scale bars described in the caption are unclear. It seems A, B, C, and D are referenced, but only images A, B, and C are shown.

Response: Thank you for pointing this out. Corrected.

2. In the 4th paragraph of section 2.1, the deflection is referenced at 10% of the lever arm, while in the second paragraph of section 3.4, a deflection of 10% of the wing's length is mentioned. It is unclear if these should be the same displacements, but they seem to reference different values as the "lever arm" refers to only 1/3 of the wing's lengths. Consider rephrasing for clarity.

Response: Thank you for pointing this out. "10% of the lever arm" is correct. This has been corrected in the revised manuscript.

3. Consider adding an explanation or graphic representation (possibly to figure 3) of how camber is being quantified in this study. V_a and V_b mentioned in the caption of figure 3 refer to the volume under the wing, however this does not align with the area and units of mm^2 shown in S14-S19. It is possible that volume changes under the full wing are not linearly proportional to area changes under the 1/3 spanwise location.

Response: Thank you for your suggestion. We have addressed this concern in our revised manuscript. Specifically, we have provided additional commentary on how camber is quantified in our study. We would like to clarify that for models of the basal complex only, we measured the camber by calculating the area below the free end of the models, as it was easily accessible. However, for complete wing models, we utilized the volume below the wing models to characterize camber.

4. Figure 4.B should be referred to as a CAD rendering of the 3D-printed part, unless it is actually a photo of the physical part.

Response: This is a 3D printed part. We have now added a video of the part to the supplementary material (Video S5).

5. The final sentence of paragraph 1 in section 3.1 implies that the specimens studied were gathered in 2016 and 1997, which is contradicted in the following sentence claiming that only “fresh” wings were used. Please correct these errors or explain specifically what the specimens from past decades were used for.

Response: This is indeed correct. This study was done in 2016 and the authors could just prepare the data for publication. We used only the specimens collected in 2016 for mechanical testing and the old specimens were used only for microscopy investigations.

6. In section 3.4, the phrase “fix the wings at their joints” implies that the wings were removed and reattached at the joint. Consider rephrasing this sentence to explain that the bee wax was used to immobilize the wing-hinge joint.

Response: Thank you for the suggestion. Corrected.

7. Validation of the FEA model in section 3.4 should be included, possibly referencing to figure S12. If the camber area under the deformed wing (S20) was gathered experimentally and can be validated against, this would significantly strengthen the reader’s trust in the modeled results.

Response: Thank you for your valuable feedback. We appreciate your suggestion and have made improvements to the manuscript in response. Specifically, we have included additional commentaries on the validation study conducted for the FEA model in section 3.4.

To validate the FEA model, we compared the force-displacement curves obtained from experiments conducted on intact wings with their respective simulations. This comparison allowed us to assess the accuracy of the model's predictions. We measured the average stiffness of both the wings and their corresponding FEA models from the two sets of curves. Our validation results revealed a difference of ~12% between the measured stiffness values. While this discrepancy indicates that further refinement is possible, it is worth noting that the model's overall performance was quite promising.

Regarding the suggestion to gather experimental data on the camber under the deformed wing and validate it against the FEA model, we acknowledge its potential to enhance the reader's trust in results. However, we would like to highlight the challenges associated with characterizing the wing's camber during experiments. This task requires significant computational work and poses practical difficulties in accurately capturing and quantifying the camber.

Nonetheless, we believe that the comparison of force-displacement curves provides valuable insights into the model's performance and offers a strong basis for assessing its reliability. We have thoroughly discussed these validation procedures and their results in the revised manuscript to ensure transparency and to instil confidence in the readers regarding the accuracy of our FEA model.

**The following comments on the FEA model could be addressed through an appendix/supplemental section to describe the model in greater detail. As this model was used to find the major trends of the parameter study, more details of the FEA model should be explained.
**

Response: We agree with the reviewer and have done our best to add more details of FEA to our revised manuscript. Please, find our point-by-point answers to your comments below.

8. In section 3.4, a thickness gradient is mentioned in the list of structural features considered in the FEA model. Does this refer to membrane thickness, vein thickness, or something else? How was this measured and applied to the model? Vein thickness is later described as constant, and a thickness distribution of the shell elements is not mentioned if it was used to represent membrane thickness gradients.

Response: Thank you for your question. In section 3.4, the thickness gradient we refer to includes changes in both the thickness of the membranes and the costal vein. We used data reported by Jongerius & Lentink (2010) to develop this gradient. For simplicity, we assumed that other veins in the wing have a constant thickness throughout. We apologize for any confusion caused by our previous lack of detail regarding this. In the revised manuscript, we have added more information about the values for each thickness and how they were applied to the FEA model in the Methods section.

9. It is mentioned that resilin patches and resilin joints were modelled, however this was not explained. Perhaps this was through connections of different modeled materials in the assembly?

Response: That's correct. This has now been explained in Methods.

10. The application of boundary conditions in the FEA model is unclear.

Response: Thank you for your comment. This has now been explained in Methods.

11. How was the discretization/mesh verified for this FEA model?

Response: In this study, we developed finite element models to simulate the mechanical behaviour of insect wings, which consist of veins and membranes. To model the veins, we selected two-node beam elements (B31), which are commonly used for modelling slender structures such as beams or trusses. For the membranes, we used general-purpose shell elements (S4R), which can accurately represent the behaviour of thin shells like wing membranes. Our selection of these elements was based on a combination of our previous experience in modelling insect wings and a consideration of the specific dimensions of the veins and membranes. We evaluated the suitability of these elements through a mesh convergence analysis, where we successively refined the mesh size until the results were independent of the mesh. Specifically, we selected the mesh size at which the results no longer changed significantly as the appropriate mesh size for our models. By performing this mesh convergence analysis, we ensured that our finite element models of insect wings were appropriately discretized and could produce accurate and reliable results. We have added more commentary on this in the revised manuscript (in Methods).

Overall, the paper presents novel work on the basal complex, and the quantitative study through physical experimentation and computational modeling was very well performed and is an interesting contribution to the field. More information about the particular setup of the FEA model should be included either in the body or the text or in an appendix of some sort. The 3D printed bioinspired wind-turbine structure is an interesting application of the work, however this section of the paper seems to be explained in less detail (this is not a major problem, but further discussion and quantitative evaluation of the proposed structure might strengthen this section of the paper for some readers if the authors deem such an addition appropriate).

Response: We sincerely appreciate the reviewer's valuable feedback on our manuscript. We are thrilled by their positive assessment, acknowledging our work as "novel" and highlighting the successful execution of the quantitative study through physical experimentation and computational modelling, which they described as "very well performed" and an "interesting contribution to the field." In response to their suggestion, we have taken great care to enhance the manuscript by providing additional details on finite element analysis, both in the main text and the supplementary materials. Furthermore, we have dedicated more attention to the section discussing the 3D printed bioinspired wind-turbine structure, addressing the need for a more thorough explanation. We believe that these revisions have strengthened the paper and offered a more comprehensive understanding of our research.

REVIEWERS' COMMENTS:

Reviewer #2 (Remarks to the Author):

As mentioned in the review comments for the original version of the manuscript, I found this study into the mechanics of the basal complex to be a very interesting read. The study is novel and the investigation performed was thorough with well documented findings and intriguing suggestions for continued work.

The revised version of the manuscript adequately addresses all of the comments that I had made previously. I do not think any additional changes are necessary.

I look forward to seeing this article in a future issue of Communications Biology!